# Drought and UV Radiation Stress Tolerance in Rice Is Improved by Overaccumulation of Non-Enzymatic Antioxidant Flavonoids

**DOI:** 10.3390/antiox11050917

**Published:** 2022-05-06

**Authors:** Rahmatullah Jan, Muhammad-Aaqil Khan, Sajjad Asaf, Muhammad Waqas, Jae-Ryoung Park, Saleem Asif, Nari Kim, In-Jung Lee, Kyung-Min Kim

**Affiliations:** 1Division of Plant Biosciences, Department of Applied Biosciences, College of Agriculture and Life Science, Kyungpook National University, 80 Dahak-ro, Buk-gu, Daegu 41566, Korea; rahmat2021@knu.ac.kr (R.J.); aqil_bacha@knu.ac.kr (M.-A.K.); icd0192@korea.kr (J.-R.P.); saleemasif10@knu.ac.kr (S.A.); jennynari@knu.ac.kr (N.K.); ijlee@knu.ac.kr (I.-J.L.); 2Costal Agriculture Research Institute, Kyungpook National University, 80 Dahak-ro, Buk-gu, Daegu 41566, Korea; 3Natural and Medical Science Research Center, University of Nizwa, Nizwa 616, Oman; sajjadasaf@unizwa.edu.om; 4Department of Botany, Garden Campus, Abdul Wali Khan University, Mardan 23200, Pakistan; lubnabilal68@gmail.com; 5Department of Agriculture Extension, Government of Khyber Pakhtunkhwa, Mardan 23200, Pakistan; agranomist89@yahoo.com; 6Crop Breeding Division, National Institute of Crop Science, Rural Development Administration, Wanju 55365, Korea

**Keywords:** drought and UV radiation stress, flavonoids content, antioxidant content, salicylic acid, genes expression

## Abstract

Drought and ultraviolet radiation (UV radiation) are the coexisting environmental factors that negatively affect plant growth and development via oxidative damage. Flavonoids are reactive, scavenging oxygen species (ROS) and UV radiation-absorbing compounds generated under stress conditions. We investigated the biosynthesis of kaempferol and quercetin in wild and flavanone 3-hydroxylase (F3H) overexpresser rice plants when drought and UV radiation stress were imposed individually and together. Phenotypic variation indicated that both kinds of stress highly reduced rice plant growth parameters in wild plants as compared to transgenic plants. When combined, the stressors adversely affected rice plant growth parameters more than when they were imposed individually. Overaccumulation of kaempferol and quercetin in transgenic plants demonstrated that both flavonoids were crucial for enhanced tolerance to such stresses. Oxidative activity assays showed that kaempferol and quercetin overaccumulation with strong non-enzymatic antioxidant activity mitigated the accumulation of ROS under drought and UV radiation stress. Lower contents of salicylic acid (SA) in transgenic plants indicated that flavonoid accumulation reduced stress, which led to the accumulation of low levels of SA. Transcriptional regulation of the dehydrin (DHN) and ultraviolet-B resistance 8 (UVR8) genes showed significant increases in transgenic plants compared to wild plants under stress. Taken together, these results confirm the usefulness of kaempferol and quercetin in enhancing tolerance to both drought and UV radiation stress.

## 1. Introduction

Over the last few decades, anthropogenic pollutants have significantly reduced stratospheric ozone, which could lead to a considerable increase in UV radiation [1]. This is a distressing situation from the standpoint of environmental safety. Living organisms are naked to UV radiation and absorb it to different degrees depending on the angle of the light and thickness of the ozone layer. The solar UV radiation consists of UV-A radiation with wavelength range from 315–400 nm, UV-B radiation with wavelength range from 280–315 nm, and UV-C radiation with wavelength below 280 nm [2], while that above 400 to 700 nm is photosynthetically active radiation (PAR) [3]. Previous reports show that an optimum amount of UV-B radiation induces some photomorphogenic responses, such as morphology, physiology, and synthesis of secondary metabolites [4]. UV radiation coupled with various other environmental factors, such as temperature, water regime, and nutrient status, can also weaken plant defenses [5]. UV radiation is one of the main sources of oxidative stress and can lead plants to accumulate UV radiation absorbing chemicals, enhance leaf thickness, and produce trichomes [6,7]. In response to UV radiation damage, plants also regulate pathways that are involved in DNA damage [8]. Numerous studies have looked at particular genes linked to UV radiation resistance [9,10,11,12]. Researchers have identified a UV-B-specific regulatory pathway that controls the genes related to UV radiation protection, which involves more specifically the UVR8 gene identified in *A. thaliana* [11,13]. Additional researchers demonstrated that UV radiation induces the accumulation of flavonoids, which provides a protective screen in the epidermal tissues of plants in association with other phenolic compounds [14]. It is well-known that UV radiation induces various genes involved in the synthesis of flavonoid pathways, such as the chalcone synthase (CHS), chalcone isomerase (CHI), flavanone 3-hydroxylase (F3H), and flavonol synthase (FLS) pathways [15,16,17]. Ref. [18] found that the mutant of UVR8 fails to induce a very key gene (chalcone synthase CHS) of the flavonoid pathway in response to UV radiation, resulting in a much-reduced quantity of flavonoids. Although the physiological role of specific flavonoids is still not well understood, it is known that kaempferol and quercetin increase under UV radiation exposure [19,20]. However, the derivatives of both flavonoids accumulate under UV stress and act as UV protectants and scavenging reactive oxygen species (ROS) [21].

Drought is one of the most common high UV radiation coexisting environmental factors that reduce agricultural production and quality [22]. Drought stress can cause various morphological, physiological, and biochemical alterations in plants, such as reduced growth, reduced stem elongation, stomatal variation, and leaf expansion [23]. Drought stress also increases hydrogen peroxide (H_2_O_2_), ROS, and superoxides, which cause protein, DNA, lipid, nucleic acid, and carbohydrate damage in rice [24,25]. ROS are regarded as by-products of plant aerobic metabolism and are generated in several cellular compartments, such as chloroplasts, mitochondria, and peroxisomes. ROS not only causes DNA damage and cell death but also act as signaling molecule. Therefore, ROS have a dual role in vivo depending on different levels of reactivity, sites of production, and potential to cross biological membranes [26]. Environmental stresses such as drought and UV radiation cause stomatal closure, leading to insufficient intercellular CO_2_ concentration, which favors the formation of ^1^O_2_ [27]. The O_2_^•−^ produced in PSI reacts with superoxide dismutase (SOD) and converts into H_2_O_2_ [28]. This single oxygen causes severe damages to both photosystems, PSI and PSII, and puts the entire photosynthetic machinery in jeopardy. Due to this damage, rice plants have adopted several strategies to cope with drought stress, including reducing life span, plasticity in development, increasing water uptake capacity, and reducing loss of water via stomatal adjustment [29]. Besides these strategies, rice plants also cope with oxidative stress through antioxidant systems, including accumulation of peroxidase, superoxide dismutase, glutathione peroxidase (GPX), polyphenol oxidase (POD), and malondialdehyde (MDA) [30,31]. Transcriptional regulation is an important strategy of plants to enhance tolerance to drought stress via strict control of physiological and biochemical responses [32]. Dehydrins (DHN) are the important hydrophilic, thermostable proteins belonging to the late embryogenic abundant (LEA) family, associated with protective function [33]. Overexpression of the OsDHN1 gene in rice enhances drought and salt stress tolerance, while the same gene in tomato confers cold and drought stress [34,35]. The expression of the DHN gene enhances accumulation of dehydrins in response to stressful conditions; these contain reactive amino acid residues and act as ROS scavengers [36]. Besides DHN, some flavonoid biosynthesis-related genes in rice plants are also regulated in response to drought stress, which indicates that flavonoids are involved in drought stress. Recently, it was reported that flavonoid contents are reduced in drought-susceptible plants due to downregulation of key genes of the flavonoid biosynthesis pathway, such as CHI in rice and dihydroflavonol-4-reductase (DFR), but are increased in tolerant plants [37,38]. This information predicts the positive role of flavonoids in mitigating drought stress and is attracting great attention from researchers to evaluate the protective mechanism of flavonoids in drought stress tolerance.

Drought and UV radiation have been studied individually, but there is very limited information about the effects of drought and UV radiation combined in creating stress on rice plants. However, a recent study was conducted on halo and UV-B radiation priming seeds of rice exposed to drought stress [39]. Additive and detrimental effects of drought and UV radiation have been investigated in several plants. *Populus cathayana* showed reduced plant height and leaf area, and *Hippophae rhamnoides* showed reduced total biomass, while *Glycine max* did not show any additive effect in response to both drought and UV radiation [40]. Some researchers also reported antagonistic effects of drought and UV radiation on conifers and European heathland species [40]. The effects of flavonoids have been studied in response to drought and UV radiation stress individually in rice plant but not in combination. Our study hypothesized that induction of some non-enzymatic antioxidant flavonoids enhances tolerance to individual as well as combined drought and UV radiation stress in the rice plant. The aim of our study is to evaluate the role of flavonoids in rice plants in response to drought and UV radiation stress individually and when they are combined. We focused on flavonoid accumulation, transcriptional regulation of drought and UV responsive genes, and antioxidant and hormonal responses in response to drought and UV radiation combined stress, revealed by overexpression of the flavanone 3-hydroxylase (F3H) gene.

## 2. Materials and Methods

### 2.1. Plant Material, Growth Conditions, and Phenotypic Evaluation

We used the rice (*Oryza sativa* L.) seeds of the transgenic (OxF3H) and non-transgenic Nogdong cultivars provided by the Plant Molecular Breeding Laboratory, Kyungpook National University, Korea [41]. Rice seeds were sterilized with fungicides overnight and then washed with double distilled water three times. Next, the rice seeds were soaked in water for four days in an incubator in the dark condition at 32 °C, with the water changed after each 24 h as previously reported by [41]. After soaking and successful sprouting, rice seeds were transferred to autoclaved soil and kept in the dark again for three days. After growth, the three week rice seedlings were exposed to light and kept in a greenhouse for further experimentation.

### 2.2. Experimental Design

We selected in total seven groups of rice plants. Among them one was control group; three were non-transgenic treated, and three were transgenic treated group. One group of non-transgenic treated and one group of transgenic treated were exposed only to drought and were named WT+D and OxF3H+D, respectively. Another group of non-transgenic- and transgenic-treated rice plants were exposed only to UV-B radiation and were named WT+UV and OxF3H+UV, respectively, while the remaining each group of non-transgenic- and transgenic-treated rice plants were exposed to drought and UV radiation combined and were named WT+D+UV and OxF3H+D+UV, respectively. The flowchart of the experimental setup is presented in Figure 1. For drought stress, rice seedling were screened at 5%, 10%, and 15% of PEG 6000 according to [42] (Appendix A). On the basis of screening, we applied 1.5 L of 10% polyethylene glycol 6000 (PEG 6000; a product of Sigma-Aldrich, Seoul, Korea) [43]. For UV radiation stress, we installed two UV-B lamp tubes 26 W (wavelength 280–315 nm, product of Dream Ocean Korea) in the growth chamber. The UV-B lamps were fixed 60 cm above the plants. To filter UVC (280 nm) irradiation, cellulose diacetate film of 0.125 mm thickness was used (Lucky Films Co., Ltd., Shanghai, China). The intensity of UVB and UVA wavelengths was measured using a Spectroradiometer 320 (1 nm resolution, Instrument System GmbH, Munich, Germany). The intensities of UVB and UVA were 0.68 and 0.27 Wm^−2^, respectively. The rice seedling after three weeks of growth in the control environment, were exposed to drought stress and UV-B radiation for 7 h per day [44]. The temperature and humidity were kept at 28 °C, −30 °C, and 60%, respectively, for all the rice plants, and light durations were kept at 16/8 h for the control and drought stress plants only. Each group of plants was first grown for three weeks in 50-hole black trays in the greenhouse and then transferred for treatment into the growth chamber except for the control plants, which were kept in a separate growth chamber with optimum growth conditions.

### 2.3. RNA Isolation and qRT-PCR

To determine the expression level of the F3H, DHN, and UVR8 genes, we collected three leaves randomly from each group of rice plants after 0, 6, 12, and 24 h of exposure to stress. The leaves were frozen in liquid nitrogen and stored at −80 °C for further analysis. Total RNA was extracted, cDNA was synthesized, and qRT-PCR was performed following [41]. In brief, RNA was extracted using RNeasy Plant Mini Kits (50) Qiagen, cDNA was synthesized using qPCRBIO kits, and qRT-PCR was performed using qPCRBIO SYBR Green kits. Primer sequence and accession number of each gene are shown in Appendix A. About 20 µL of reaction was initiated using 10 µL SYBR green, 7 µL ddH_2_O, 1 µL template DNA, and 1 µL of each primer. The reaction was incubated at 95 °C for 2 min, followed by 35 cycles at 94 °C for ten seconds, and 60 °C and 72 °C for ten and forty seconds, respectively. Three independent reactions were performed for each sample on an Eco Real-Time (Illumina, Singapore) machine. Actin was used as an internal reference gene and relative expression levels were quantified using the 2^−ΔΔCt^ method.

### 2.4. Quantification of Endogenous Salicylic Acid

Freeze-dried rice leaves were ground into fine powder in liquid nitrogen, and a 0.2 g sample was homogenized in 90% ethanol and 100% methanol as previously described by [45]. The homogenate was centrifuged at 10,000× *g* for 15 min. The supernatant was collected, and both ethanol and methanol of the supernatant were dried in a vacuum centrifuge. The pellet was suspended in 5% trichloroacetic acid (3 mL). The supernatant was further mixed with ethyl acetate/cyclopentane/isopropanol (49.5:49.5:1, *v*/*v*/*v*), and the uppermost organic layer was collected in a 4 mL vial and dried with nitrogen gas. The dry SA was again suspended in 1 mL of 70% methanol, and 25 μL of the filtered sample was subjected to high-performance liquid chromatography (HPLC) with a C18 reverse-phase HPLC column (HP hypersil ODS, particle size 5µm, pore size 120 Å Waters). The flow rate was 1.0 mL/min, and water was used as eluent as shown in Appendix A. SA was detected with a Shimadzu fluorescence detector (Shimadzu RF-10AXL) with excitation and emission monitored at 305 and 365 nm, respectively [46]. The quantity of SA was calculated according to peak value for authentic standards.

### 2.5. Detection of H_2_O_2_ and Cell Death

H_2_O_2_ was detected visually in the rice leaves after three days of treatment by using diaminobenzidine (DAB) as a substrate, following [47]. Rice leaves from each group of treatment were incubated in DAB (1 mg mL^−1^) for 24 h at 27 °C as described by Chao et al. (2010). Leaves were then de-stained with 95% ethanol by boiling for 30 min. Ethanol treatment decolorized the leaves’ chlorophyll, and the remaining color was the brown polymerization product produced by DAB with H_2_O_2_. After boiling, the leaves were cooled at room temperature in ethanol to detect the brown spots. Cell death was detected by trypan blue staining following the protocol of [48]. Rice leaves and roots were collected randomly from each treatment after exposure to the stress condition. Briefly, leaves and roots were incubated in a trypan blue mixture (2.5 mg/mL trypan blue, 25% *w*/*v* lactic acid, 23% phenol, and 25% glycerol) at room temperature for 4 h. After proper staining, samples were de-stained with 95% ethanol, keeping them at room temperature overnight while shaking. After de-staining, samples were washed 3–5 times with distilled water for taking pictures.

### 2.6. Glutathione Peroxidase Activity and Measurement of Malondialdehyde Contents

Glutathione peroxidase (GPX) activity and malondialdehyde (MDA) contents were determined using the Glutathione Peroxidase Cellular Activity Assay Kit (Sigma, Seoul, Korea) and Lipid Peroxidation (MDA) Assay Kit (Sigma), respectively, according to the manufacturer’s manuals as described in [45].

For GPX detection, 100 mg deep-frozen rice leaves were ground in liquid nitrogen, homogenized in 3 mL 5% trichloroacetic acid (TCA), and centrifuged at 15,000× *g* for 15 min as described [49]. The extract was used for further analysis and blank, positive control, and sample reactions were performed. The kit provided the glutathione peroxidase assay buffer, NADPH assay reagent, and Luperox 70% tert-butyl-hydroperoxide (TBH70X). According to the user manual, 1 vial of NADPH assay reagent was reconstituted in 1.25 mL ddH_2_O. To prepare 30 mM tert-butyl-hydroperoxide, 21.5 µL Luperox TBH70X was diluted in ddH_2_O to 5 mL. All reactions were prepared at a 1 mL volume, and then, 250 µL was placed in a 96-well microplate. The reaction was initiated by adding 10 µL 30 mM tert-butyl hydroperoxide. The decrease in absorbance at 340 nm was calculated using wavelength 340 nm, initial delay 15 s, interval 10 s, and number of readings 6. The GPX activity was calculated in units/mL using the following formula:

Activity per extract (µmol/min/mL = Units/mL)
ΔA340/6.22 × DF/V
where

ΔA340 = A340/min (blank) − A340/min (sample);

6.22 = εmM for NADPH;

DF = dilution factor of sample before adding to reaction;

V = sample volume in mL.

For MDA contents measurement, the kit provided the MDA lysis buffer, phospho-tungstic acid, BHT 100X, TBA, and MDA standard 4.17 M. The TBA solution was reconstituted by adding 7.5 mL glacial acetic acid (not provided), the volume was adjusted to 25 mL by adding ddH2O, and the solution was sonicated. To prepare 2 mM standard MDA, a 10 µL 4.17 M MDA solution was diluted with 407 µL ddH2O, and then, the 100 µL diluted solution was added to 900 µL ddH2O to prepare a 0.2 mM MDA solution. Subsequently, 0, 2, 4, 6, 8, and 10 µL 0.2 mM MDA standard solution was added into a 96-well microplate, and 0 (blank) 0.4, 0.8, 1.2, 1.6, and 2.0 µL standards were prepared. Thereafter, ddH2O was added to each tube to reach a volume of 200 µL. Samples were prepared by homogenizing 10 mg tissue with 300 µL MDA lysis buffer containing 3 µL BHT on ice. The samples were centrifuged for 10 min at 13,000× *g,* and the debris was discarded. Then, 200 µL of each sample was placed in a 1 mL tube to which 600 µL TBA was added, incubated for 1 h at 95 °C, and cooled by keeping on ice for 10 min. Finally, 200 µL blank and samples were pipetted into a 96-well microplate, and absorbance was analyzed at 532 nm. The reaction was run in three technical replicates, the MDA contents were calculated in µg/g, and data were calculated using the following formula:Sa/Sv × D = C

Sa = amount of MDA in unknown sample (nmole);

Sv = sample volume added into each well (mL);

D = sample dilution factor;

C = concentration of MDA in the sample.

### 2.7. Isolation, Quantification, and Detection of Flavonoids

To quantify kaempferol, quercetin, and naringenin, rice leaves were collected randomly from each treatment group following the protocol described by [50], with slight modifications. About 3 g of fresh leaves were ground in liquid nitrogen into fine powder and homogenized with 30 mL methanol H_2_O and a HCl mixture (MeOH:H_2_O:HCl = 79:20:1, *v*/*v*/*v*). Initially the homogenized sample were sonicated for 20 min and then directly subjected to incubation at 27 °C for 24 h in a shaking incubator. The crude extracts were filtered, and the filtrate was diluted using a rotary evaporator at 30 °C. The extract was dried in the heating block at 60 °C for overnight. The dried crude extract was dissolved in 1 mL HPLC grade ethanol, and 250 µL was used for spectrophotometry. The absorbance of kaempferol, quercetin, and naringenin was measured at 265 nm, 370 nm, and 285 nm, respectively, using spectrophotometer (UV-2450; Shimadzu, Tokyo, Japan). Reference standards for spectrophotometry were prepared by dissolving 1 mg of each standard sample in 1 mL ethanol. To prepare standard curves, appropriate volume of each stock solution was transferred and diluted to final concentration of 0, 2, 4, 6, 8, and 10 µg/mL. All samples were analyzed in triplicate.

To visualize the accumulation of flavonoids in the rice tissue, we collected fresh leaves and roots from control, WT+D+UV, and OxF3H+D+UV plants. Diphenylboric acid-2-aminoethyl ester (DPBA) staining was used for detection of flavonoids in the tissue as described in Jan et al. (2020). DPBA staining solutions were prepared by mixing 0.25 g (0.25%) DPBA and 200 µL Triton X-100 (0.02% *v*/*v*) in ddH_2_O up to a final volume of 100 mL. The samples were then incubated in 0.25% staining solution in a vacuum for 5 min. After staining, the samples were mounted on microscope slides for confocal microscopy. A confocal laser scanning microscope (CLSM; Carl Zeiss LSM700, White Plains, NY, USA) was used to detect the fluorescence of flavonoids [51]. An FITC filter (suppression LP 488 nm) was used for visualizing Kr (green), and an R-PE filter (suppression LP 488 nm) and rhodamine filter (suppression LP 555 nm) were used for visualizing Qu (orange) and naringenin (red), respectively.

### 2.8. Sodium, Potassium, and Calcium Accumulation

To determine Na+, K+, and Ca contents, we collected fresh shoots of all the treatment groups of rice plant. A sample of about 0.05 g was crushed in liquid nitrogen and homogenized in 7 mL of 65% NHO_3_ and 1 mL of 30% H_2_O_2_ and kept in a microwave (180 °C; 20 min) and cooled for 40 min as described by [52]. The obtained solvent was quantified by using inductively coupled plasma mass spectrometry (9ICP-MS; Optima 7900DV, Perkin-Elmer, Waltham, MA, USA).

### 2.9. Chlorophyll Content

The chlorophyll contents were measured using a soil plant analysis development (SPAD-502plus from Konica Minolta Sensing, Seoul, Korea) chlorophyll meter using randomly selected rice leaves in triplicate after one week of continuous exposure to the stress condition [45].

### 2.10. Statistical Analysis

All experiments were performed in triplicate, and the data from each replicate were pooled. Data were analyzed using two-way ANOVA with Bonferroni post hoc tests (* shows *p* < 0.05, ** shows *p* < 0.01, and *** shows *p* < 0.001 significant difference) and Duncan’s Multiple Range test. A completely randomized design was used to compare the mean values of different treatments. Data were graphically plotted, and statistical analyses were performed using the GraphPad Prism software (version 5.01, GraphPad, San Diego, CA, USA) and Statistical Analysis System (SAS 64 bit, developed by North Carolina State University, Raleigh, NC, USA).

## 3. Results

### 3.1. Plant Growth under UV-B Radiation and Drought Stress

In this experiment, we evaluated various growth parameters of rice plant in response to combined and individual stresses as shown in Figure 2. The data indicated that rice plant responses to combined stress were significantly different from the responses displayed in response to individual stress. Moreover, the response of the transgenic line was different from the wild-type plants. Combined drought and UV-B radiation had an interactive, negative effect on shoot length of Wt+D+UV and OxF3H+D+UV plants and reduced the shoot length to 26% and 16%, respectively, compared with control plants (Figure 2A). However, the individual drought and UV-B radiation stressors significantly reduced shoot length in wild plant, while transgenic plants showed only 2% reduction compared with control plants. The root length was reduced in Wt+UV and Wt+D+UV plants to 28% and 31%, respectively, while it increased 9% in Wt+D plants (Figure 2B). In contrast, root length was increased significantly in all the transgenic plants. Although drought and UV-B radiation significantly decreased the leaf area, leaf area of Wt+D+UV was severely (42%) reduced followed by OxF3H+D+UV (36%; Figure 2C). The highest leaf-tip burn was caused by combined stress in wild-type (8.5 cm) followed by individual drought stress (6.3 cm) in OxF3H plants (Figure 2D). In Figure 2E, we document the root hair development in wild and transgenic plants exposed to different stresses. We also visualize the intensity of stress by indicating symptoms on leaves caused by different stresses (Figure 2F). The symptoms show that the stress intensity was higher in Wt+D+UV plants, followed by Wt+UV and OxF3H+UV plants.

### 3.2. OxF3H Reduces Oxidative Stress via Regulation of Glutathione Peroxidase and MDA Contents

We determined oxidative stress via glutathione peroxidase (GPX) and lipid peroxidation (MDA) and visualized it via trypan and DAB histochemical staining. The GPX activity was lower in OxF3H+UV (72.5%) and OxF3H+D+UV (72.4%) rice plants as compared to Wt+UV and Wt+D+UV rice plants (Figure 3A), which indicates that the transgenic line reduced oxidative stress. In contrast, the lipid peroxidase activity was significantly higher in wild-type plants and was reduced in transgenic plants as compared to control plants (Figure 3B). The highest increase was shown by Wt+D+UV (303%), followed by Wt+UV (226%) and Wt+D (171%) plants. In contrast, OxF3H+D+UV plants showed the highest decrease (62%), followed by OxF3H+UV (39%) and OxF3H+D (29%). This shows that the transgenic line reduces lipid peroxidation under drought and UV-B radiation stress. The oxidative stress induced by drought and UV-B radiation was further examined by subjecting the leaves to DAB and trypan staining. As shown in Figure 3C, dark stained patches appeared in Wt+D and Wt+D+UV plants, and little DAB staining appeared in OXF3H+UV and OxF3H+D+UV leaves. These observations indicate that increased staining in wild plants induced by individual UV-B radiation stress and drought and UV-B radiation combined stress was caused by a high level of generation of H_2_O_2_, which caused oxidative damage. Furthermore, cell death caused by oxidative stress was also determined in roots using trypan blue staining (Figure 3D). In transgenic plants, less trypan accumulated in OxF3H+D and OxF3H+UV as compared to OxF3H+D+UV plants, which indicates that combined stress causes more oxidative damage than individual stress. The close image of oxidative damage detected with DAB staining is shown in Appendix A.

### 3.3. Overexpression of F3H Regulates Drought and UV-B Radiation Responsive Genes

Our results showed that the expression of OsF3H was significantly higher in transgenic rice plants than the wild rice plants under both the stresses (Figure 4A). The highest increase was reported as 711%, 480%, 471%, and 463%, respectively, in OxF3H+D+UV (24 h) plants, followed by OxF3H+UV (12 h), OxF3H+D+UV (12 h), and OxF3H+UV (24 h) plants, respectively. The drought-stress-responsive gene (DHN) in wild plants was expressed significantly (*p* ≤ 0.01) in Wt+D plants after 12 h and in Wt+D+UV plants after 24 h, and the level of expression was increased to 393% and 348%, respectively (Figure 4B). However, in transgenic plants, DHN was significantly (*p* ≤ 0.001) expressed in OxF3F+D plants after 12 h and 24 h, and the level of expression was 653% and 523%, respectively. In OxF3H+D+UV plants, the DHN was expressed 437%, 571%, and 387% more compared to control plants after 6 h, 12 h, and 24 h of exposure to combined stress, respectively. The UVR8 gene was induced in plants under the UV-B radiation stress and drought and UV-B radiation combined stress in both wild and transgenic plants (Figure 4C). In wild plants, the expression level of UVR8 was significantly (*p* ≤ 0.001) increased 155% and 245% in Wt+UV after 12 h and 24 h, respectively, and was increased in Wt+D+UV 146% after 24 h of exposure to stress as compared to control plants. In transgenic plants, the UVR8 expression was significantly (*p* ≤ 0.001) induced in OxF3H+UV and OxF3H+D+UV plants after 12 h and 24 h of stress exposure. The level of expression was 505% and 571% in OxF3H+D plants after 12 h and 24 h and 283% and 362% in OxF3H+D+UV plants after 12 h and 24 h, respectively, as compared to control plants.

### 3.4. OxF3H Induces Flavonoid Biosynthesis in Response to Drought and UV-B Radiation Stress

Kaempferol, quercetin, and naringenin were detected in rice leaf and root using CLSM as shown in Figure 5, while the quantitative determination is shown in Figure 6. Different patterns of kaempferol, quercetin, and naringenin accumulation were determined in control, Wt+D+UV, and OxF3H+D+UV plants after 7 h of continuous exposure to combined stress. Large variation in kaempferol and quercetin accumulation was detected in the leaves of Wt+D+UV and OxF3H+D+UV plants as compared to control leaves. However, in Wt+D+UV plants, large amounts of quercetin were detected, while in Oxf3H+D+UV plants, large amounts of kaempferol were detected. The amount of naringenin was higher in Wt+D+UV as compared to OxF3H+D+UV plant leaves. In contrast, OxF3H+D+UV roots showed large amounts of kaempferol but very little quercetin and naringenin. These results indicate that naringenin converts very rapidly into flavonoids in response to drought and UV-B radiation stress (see Figure 5). The close image of detection of kaempferol and quercetin is represented in Appendix A.

The quantitative results of kaempferol, quercetin, and naringenin were in line with the histochemical detection. The biosynthesis of kaempferol and quercetin was increased in both wild and transgenic rice plants under individual and combined stress as compared to control plants (Figure 6A). However, kaempferol was increased 227%, 244%, and 285% in OxF3H+D, OxF3H+UV, and OxF3H+D+UV plants, respectively. Similarly, quercetin was increased 136%, 311%, and 341% in OxF3H+D, OxF3H+UV, and OxF3H+D+UV plants, respectively (Figure 6B). Unlike kaempferol and quercetin, naringenin was reduced in the stress condition in both wild and transgenic plants. A very small reduction of naringenin was found in both Wt+D (15%) and Wt+D+UV (15%), while in OxF3H+D, OxF3H+UV, and OxF3H+D+UV, the reduction in naringenin was reduced significantly to 25%, 70%, and 91%, respectively (Figure 6C). This indicated that overexpression of OsF3H induced conversion of naringenin into kaempferol and quercetin in response to drought and UV-B radiation individual stress as well as combined stress.

### 3.5. OsF3H Regulates Salicylic Acid and Photosynthesis under Drought and UV-B Radiation Stress

The results showed that compared with control rice plants, both wild and transgenic rice plants enhanced the accumulation of SA contents (Figure 7A). However, comparing wild and transgenic plants, we found that wild plants accumulated more SA contents than transgenic plants. Wt+D+UV and Wt+UV accumulated the highest SA contents followed by Wt+D at 597%, 581%, and 260%, respectively, as compared to control plants. The highest SA contents in the transgenic plants were accumulated in OxF3H+D+UV (208%) followed by OxF3H+D+UV (102%) and OxF3H+D (41%). Although transgenic plants increased SA accumulation as compared to control plants, the increase was minor with respect to wild plants. These results determined that OsF3H overexpression reduced the accumulation of SA in drought and UV-B radiation stress.

Similarly, chlorophyll contents were significantly reduced in wild as well as transgenic rice plants, indicating that drought and UV-B radiation inhibit the rate of photosynthesis. However, comparing wild and transgenic plants, the results showed that chlorophyll contents were highly reduced in wild plants compared to transgenic plants (Figure 7B). The highest reduction was reported in both wild and transgenic plants exposed to combined stress, which indicates that combined stress inhibits photosynthesis more than does individual stress.

### 3.6. Ion Content Changes in Wild and Transgenic Plants under Drought and UV-B Radiation Stress

The Inductively Coupled Plasma Mass Spectrometric (ICP-MS) analysis of Na+, K+, and Ca+ showed different accumulations in wild and transgenic rice plants under drought and UV-B radiation stress (Figure 8). In general, both the wild and transgenic plants showed higher accumulation of Na+, K+, and Ca+ contents as compared to control plants. In particular, Na+ accumulation was significantly higher in Wt+D+UV (103%), followed by OxF3H+D+UV (52%) and Wt+UV (44%) (Figure 8A). Furthermore, the highest accumulation of K+ was found in OxF3H+D+UV (54.8%) plants, followed by OxF3H+UV (29.7%) plants, while wild-type plants showed 23.9%, 8.2%, and 8.3% increases in Wt+D+UV, Wt+UV, and Wt+D plants, respectively, as compared to control plants (Figure 8B). Additionally, the accumulation of Ca+ was significantly higher in the transgenic line exposed to individual as well as combined stress compared to wild and control plants (Figure 8C). OxF3H+D+UV accumulated 34.4%, OxF3H+UV accumulated 31.4%, and OxF3H+D accumulated 12.9% Ca+ as compared to control plants, while Wt+D+UV accumulated 21.4% Ca+ as compared to control plants. However, Wt+D and Wt+UV plants showed accumulation of very similar amounts of Ca+ as control plants. The regulation of Na+, K+, and Ca+ contents are involved in plant stress reduction; therefore, these results indicate that OsF3H transgenic lines enhanced the accumulation of these ions under both the stresses.

## 4. Discussion

In the current study, drought and UV-B radiation combined stress caused huge depletions in rice growth vigor (Figure 2). The transgenic rice plants showed enhanced growth and developed root hairs and reduced leaf-tip burn. UV-B radiation greatly influence the growth vigor compared to drought stress. It is reported that UV-B radiation stress resulted in a reduced growth rate and reduced leaf area in pea and wheat plants as compared to drought stress [7,23]. Ref. [23] reported that drought-stress inhibits plant growth rate, stem elongation, leaf area, and stomatal movement. Drought stress reduces cell division and cell expansion rate, which lead to reduced internal node elongation, leaf area, and total dry weight [53]. Flavonoids are prominent metabolites that play a key role in plant responses to biotic and abiotic stress and significantly enhance growth under stress conditions [54,55,56]. They are multifunctional and act as non-enzymatic antioxidants, developmental regulators, and photo-protectants [57]. Both drought and UV-B radiation stress caused oxidative damage in leaves as well as in roots of rice plants (Figure 3). Due to greater tolerance of transgenic rice plants to UV-B radiation, it is believed that kaempferol and quercetin are produced in response to drought and UV-B radiation stress to reduce the damaging effect of both the stresses. It is reported that some plant species enhance their antioxidant system to protect themselves from UV-B radiations and accumulate UV-absorbing compounds [58]. Therefore, understanding the flavonoid modification is necessary to develop UV-B radiation tolerance in plants through molecular approaches. The quantitative analysis showed that higher kaempferol and quercetin was accumulated in transgenic plants exposed to combined stress followed by transgenic plants exposed to individual stress (Figure 6). A recent study on *Arabidopsis thaliana* has also confirmed by transcriptomic evidence the enhancement of flavonol metabolism under drought conditions [59]. Mostly UV-B radiation inhibit plant physiological and metabolic process by affecting the photosynthetic apparatus, resulting in decreased efficiency of photosystem II by accelerating the chlorophyll degradation [60]. However, plants develop a wide range of defensive strategies, such as DNA repair and synthesis of UV-B-absorbing compounds, such as flavonoids, anthocyanin, and carotenoids, to reduce the damage cause by UV-B radiation [61]. Comparing the accumulation of kaempferol and quercetin with DAB and trypan blue staining results, kaempferol and quercetin clearly reduced oxidative damage by scavenging the ROS generated during the stress condition. Our results were in line with the results reported by [62], which found that both kaempferol and quercetin possess high ROS scavenging activity in *Camellia sinensis* under cold stress. This suggests that the plants that accumulated more flavonoids showed less oxidative damage and vice versa. A previous report shows that *Salix myrsinifolia* increases UV-B radiation-absorbing compounds when the UV-B radiation increases [63]. Ref. [64] reported that flavonoid biosynthesis protects the wheat plant from UV-B radiation damage. Previous authors reported that UV-B radiation stress enhanced accumulation of flavonoid contents [65], while [66] reported that UV-B radiation stress decreases flavonoid contents in *Bryum argenteum* and *Didymodon vinealis* species. Ref. [67] proposed that UV absorbing compounds reach the saturation point as UV-B levels reach the plant’s tolerance threshold so that they lose the capacity to protect from further damage. The decreased damage shown by the transgenic plants was due to the higher accumulation of kaempferol and quercetin. As it has been previously reported, flavonols such as kaempferol, quercetin, and myricetin are UV-screening pigments since they have long been considered as the most effective UV-B absorbers, thus conferring strong photo-protection [68]. In *Arabidopsis* mutants (tt5-CHI), the absence of kaempferol and quercetin increase chloroplast damage [69]. In addition, kaempferol and quercetin have the potential to reduce the oxidative stress caused by drought stress via scavenging ROS. Our investigation indicated that, due to the low accumulation of kaempferol and quercetin in the wild plants, the wild-type plants shows more oxidative damage as compared to transgenic plants. (Figure 3 and Figure 6). This indicates that higher kaempferol and quercetin scavenge high ROS and vice versa. These results are supported by previous reports; for example, in *Arabidopsis*, kaempferol and quercetin have higher capacity for ROS scavenging than the other flavonoids [70]. Several studies reported that flavonoids are influential ROS scavengers in response to drought stress [59,71,72]. For instance, flavonoids were increased in leaves of *Cistus clusii*, a cell suspension of *Glycyrrhiza inflate*, leaves of *Brassica oleracea*, and different parts of *Leonurus cardiaca* plants when exposed to different drought stresses [73,74,75,76].

In the current study, we found that the level of glutathione peroxidase was higher in wild rice plants than transgenic plants. These results are contradictory with the accumulation of kaempferol and quercetin, as the transgenic plants accumulated higher kaempferol and quercetin than wild plants (Figure 3A and Figure 6A,B). They indicated that due to overaccumulation of kaempferol and quercetin, transgenic plants suppress the enzymatic antioxidants. They also are consistent with [77], who reported that due to non-significant differences in antioxidant enzymes, fulvic acid enhanced non-enzymatic antioxidants in scavenging the ROS generated during stress conditions. However, wild plants have lower capacities to activate the non-enzymatic antioxidants and therefore activate enzymatic antioxidant machinery. Additionally, the higher MDA contents in wild plants indicated that the application of drought and UV-B radiation stress caused greater membrane damage. Our results were consistent with [23], who found that UV-B radiation exposure produces significant membrane damage in pea and wheat plants, as assessed by lipid peroxidation and electrolytic leakage, while drought stress also increases lipid peroxidation in *Medicago truncatula* [78] and in soybean roots [79].

Further, the results indicated that the expression of F3H was not consistent in wild plants, and the level of expression was lower than in transgenic plants. These findings show that plants activate their flavonoid biosynthesis pathway up to a certain level during drought and UV-B radiation stress, but in response to continuous stress, they compromise on oxidative injuries by downregulation of the F3H gene. Previous study showed that the F3H gene in *Reaumuria soongorica* and *Vitis vinifera* plants was upregulated in response to drought stress and increased flavonoid contents, while in *Camellia sinensis*, it was downregulated in response to drought stress [80,81,82]. Several other studies also showed that flavonoid contents were increased by upregulation of the F3H gene in response to UV-B radiation. For instance, [83] reported that F3H and DFR genes were induced in several stresses, such as salt stress, UV-B radiation stress, drought, and nitrogen stress. Additionally, the overexpression of OsDHN in rice plants shows high tolerance to drought and salt stress [34], while some research found a positive correlation between DHN and cold and drought tolerance in tomato plants [35]. The DHN gene was induced in various plants under various abiotic stresses, which revealed that this protein has an important role in protection of plants during cellular dehydration. Although the function of dehydrin is not fully understood, [84] reported that the overexpression of DHN *Citrus unshiu* Marcov increased cold tolerance by reducing lipid peroxidation via directly scavenging radicals. Similarly, [85] reported that UVR8 in tomato enhances the light-absorbing component under UV-B radiation stress. Furthermore, researchers suggested that UVR8 is linked to a signaling pathway that leads to a complex series of plant responses against UV-B radiation [86]. Although the exact mechanism is still not understood and needs to be further studied, there is a link between UVR8 induction and flavonoid biosynthesis in response to UV-B radiation. As has been reported, the UVR8 mutant of *Arabidopsis* fails to induce the chalcone synthase (CHS) gene under UV-B radiation and reduces the level of flavonoid biosynthesis [87]. However, it is reported that drought regulates key genes coding for the enzymatic activity involved in flavonol biosynthesis, such as chalcone isomerase (CHI), flavonoid 3′-hydroxylase (F3′H), flavanone 3-hydroxylase (F3H), and flavonol synthase (FLS), which results in increased flavonol concentrations [81,88]. Our results indicated that both DHN and UVR8 were upregulated in transgenic plant is response to drought and UV-B radiation, respectively. This co-expression of both the genes with overexpression of the F3H gene indicates that there is a positive correlation between the F3H gene and DHN and UVR8 genes. However, any linkage of DHN and UVR8 genes with the expression of F3H is still unknown and requires additional study.

SA has been regarded as an essential plant hormone in tolerance to abiotic stress. SA has been reported as an important tool of drought-stress tolerance by enhancing plant growth and reducing lipid peroxidation and superoxide contents [89,90]. In the present study, SA accumulation was significantly increased in wild plants, while the transgenic plants showed reduced accumulation compared to wild plants (Figure 7). Our results are consistent with the study of [91], who reported the increase level of SA in *Brassica napus* specie during drought stress. These results suggest that transgenic plants reduced both the drought and UV-B radiation stress via enhanced accumulation of kaempferol and quercetin and thus reduced oxidative stress and lipid peroxidation. We believe that due to lack of tolerance to drought and UV-B radiation stress, wild plants activate the SA pathway to cope with stress conditions. SA accumulation reduces oxidative injuries via increasing ROS scavenging antioxidant enzymes and also enhances chlorophyll contents [44]. Likewise, due to tolerance to both stresses, transgenic plants showed more accumulation of chlorophyll than the wild plants, which indicates that transgenic plants enhance photosynthesis as compared to wild plants. Exogenous application of SA significantly enhances photosynthesis in UV radiation-exposed plants [83,92]. However, Figure 7 shows that transgenic plants accumulated less indigenous SA, while the chlorophyll content was higher in transgenic plants and vice versa in wild plants. Our results suggest that overaccumulation of flavonoids in transgenic plants reduces abiotic stress, which results in reduced SA content and enhanced chlorophyll content as compared to wild plants.

Our study indicated that high levels of Na+ accumulated in wild plants, which were susceptible to drought and UV-B radiation stress, while K+ and Ca+ were higher in transgenic plants, which were tolerant to both the stresses. We envisage that Na+ accumulation is linked with generation of ROS because in transgenic plants the ROS were reduced by overaccumulation of flavonoid and also showed lower levels of Na+ as compared to wild plants. It is known that increased levels of Na+ can disturb uptake of K+, which can lead to lower plant dry matter and sometimes cause plant death [93]. Furthermore, salinity increases NA+, which suppresses the level of K+, Ca+, and Mg+ in cotton plants [93]. Additionally, rutin is an important flavonoid that regulates Na+ and K+ homeostasis. Ref. [94] reported that rutin enhances salt tolerance via exclusion of Na+ and retention of Ka+ in leaf mesophyll tissue. Consistent with these results, we predicted a positive correlation of Na+, K+, and Ca+ ions with kaempferol and quercetin accumulation, but the mechanism by which kaempferol and quercetin regulates these ions still needs to be investigated.

## 5. Conclusions and Future Study

The key finding of this study is the discovery of enhanced kaempferol and quercetin biosynthesis in the OxF3H transgenic rice plants in response to both stresses. This quantitative pattern of both the flavonoids gives rise to diverse drought and UV-B radiation-responsive mechanisms, including ROS scavenging, lipid peroxidation reduction, specific gene expression regulation, hormonal regulation, and ion homeostasis. Our study showed that both the stresses induced oxidative damage by generation of ROS and membrane damage due to lipid peroxidation, which leads to reduced rice plant growth and development. This study determined that OxF3H transgenic rice line is more tolerant to drought and UV-B radiation stress than wild-type rice plants. Both kaempferol and quercetin enhanced stress tolerance by scavenging the ROS and capturing the UV-B radiation. Although the transgenic plants improved the transcriptional regulation of the DHN and UVR8 genes in stressed plants, this suggests that F3H overexpression has a beneficial relationship. The exact mechanism, however, is yet unknown and requires additional investigation.

## Figures and Tables

**Figure 1 antioxidants-11-00917-f001:**
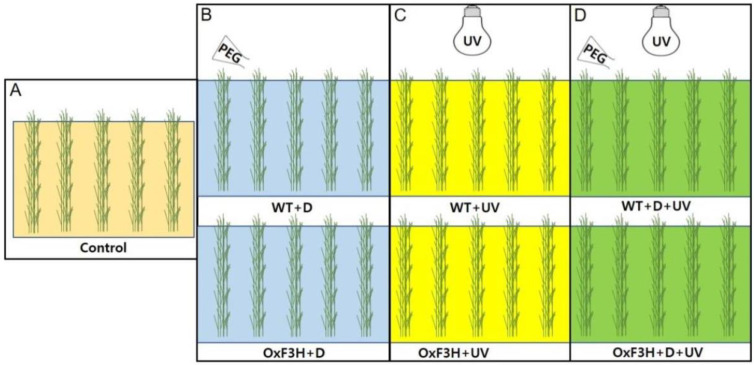
Experimental design of the growth chamber experiment, indicating the seven experimental boxes and application of drought and UV-B radiation stress. Box (**A**) shows control plants. Box (**B**) shows drought-exposed plants only; the upper picture represents wild plants (Wt+D), and the lower picture represents transgenic plants (OxF3H+D). Box (**C**) shows the plants exposed to UV-B radiation stress; the upper picture is the wild plants (Wt+Uv) exposed to UV-B radiation, and the lower picture represents transgenic plants (OxF3H+Uv) exposed to UV-B radiation stress. Box (**D**) shows plants exposed to drought and UV-B radiation combined stress; the upper picture shows wild plants (Wt+D+Uv), and the lower picture shows transgenic plants (OxF3H+D+Uv).

**Figure 2 antioxidants-11-00917-f002:**
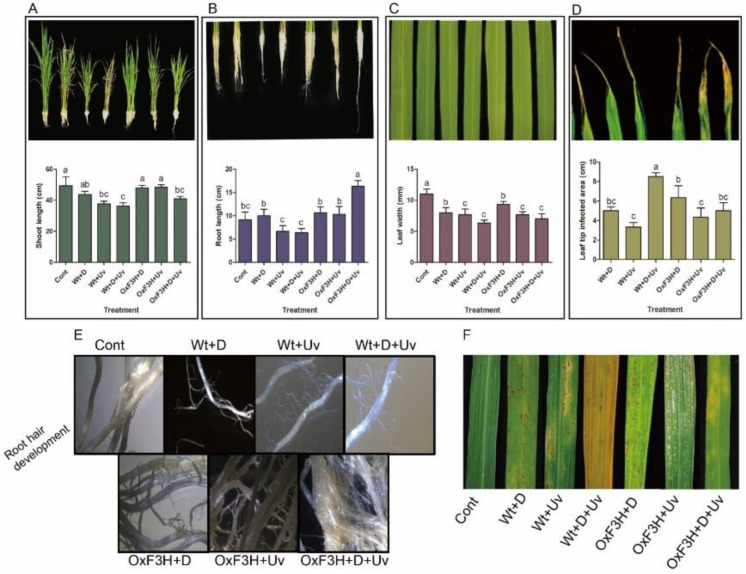
Evaluation of growth parameters in wild and transgenic rice plants under drought and UV-B radiation stress. (**A**) Shoot length, (**B**) root length, (**C**) leaf width, and (**D**) leaf tip damaged due to stress. In each box, the upper part is graphical, while the lower part is a pictorial representation of each phenotype. (**E**) The root hair development in in wild and transgenic plants and (**F**) symptoms appearing on leaves under different stresses. Each data point in (**A**–**D**) is the mean of three replicates. Error bars represent standard errors. The bars shown with different letters are significantly different from each other as evaluated by DMRT analysis.

**Figure 3 antioxidants-11-00917-f003:**
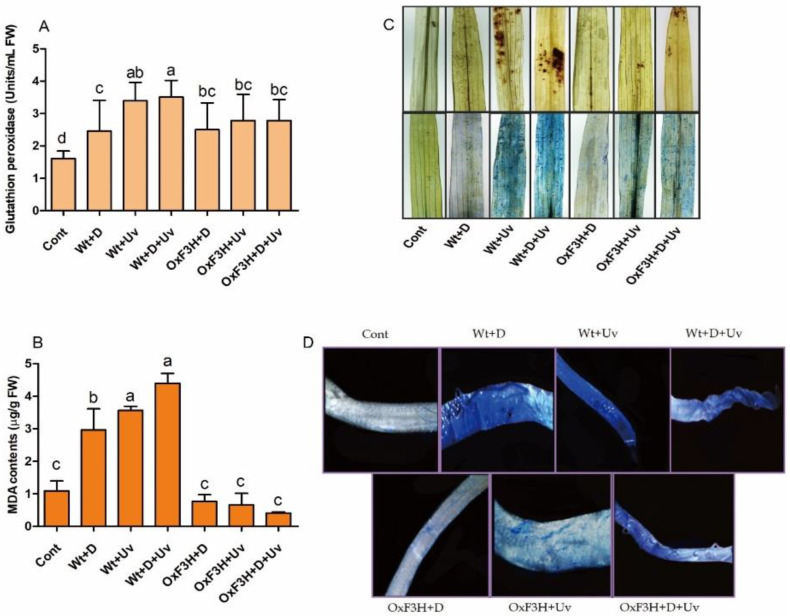
Oxidative stress analysis and glutathione peroxidase and malondialdehyde content regulation under drought and UV-B radiation stress in wild and transgenic rice plants. (**A**) Glutathione peroxidase activity and (**B**) malondialdehyde accumulation. (**C**) Oxidative damages; the upper part (**C**) indicates DAB staining, and the lower part indicates trypan blue staining. (**D**) Oxidative damages in roots. Each data point in (**A**,**B**) is the mean of three replicates. Error bars represent standard errors. The bars shown with different letters are significantly different from each other as evaluated by DMRT analysis.

**Figure 4 antioxidants-11-00917-f004:**
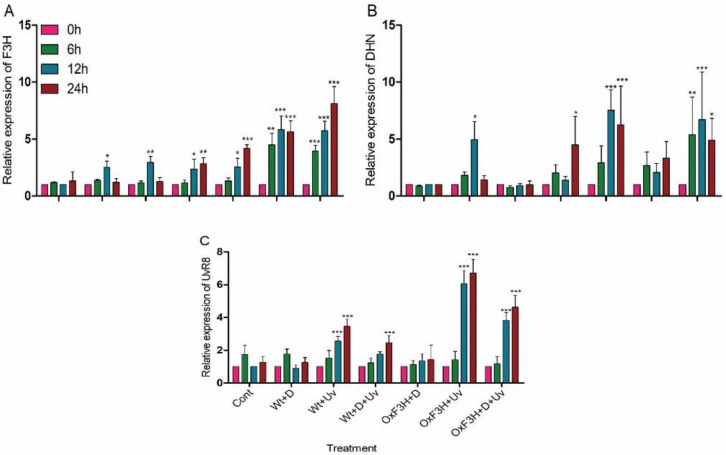
Regulation of F3H, DHN. and UVR8 gene expression under drought and UV-B radiation stress in wild and transgenic rice plants. (**A**–**C**) The relative expression of F3H, DHN, and UVR8 genes, respectively. The fold change of each gene was measured after 0 h, 6 h, 12 h, and 24 h intervals via qPCR using Actin as a reference gene. Graph bars indicate mean ± standard deviation and asterisks show a significant difference (* *p* ≤ 0.05, ** *p* ≤ 0.01, *** *p* ≤ 0.001) analyzed by two-way ANOVA with Bonferroni correction.

**Figure 5 antioxidants-11-00917-f005:**
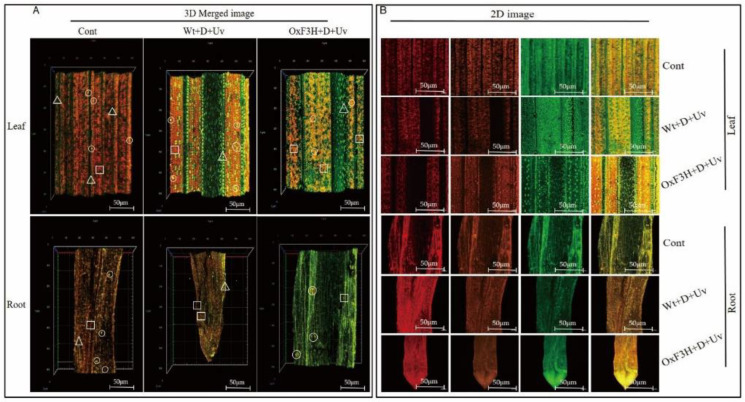
Detection of flavonoids in leaf and root under drought and UV-B radiation combined stress. Naringenin, kaempferol, and quercetin were visualized in rice plant tissue via DAPB staining using a confocal laser scanning microscope (CLSM). Box (**A**) shows a 3D image of naringenin, kaempferol, and quercetin merged in a single image. The upper images in box A are leaf images, and the lower are root images. Naringenin, kaempferol, and quercetin were detected in all the leaves and roots with different intensities. Red color shows naringenin, orange shows quercetin, and green shows kaempferol, indicated by □, O, and Δ, respectively. Box (**B**) shows 2D images of single and merged flavonoids. The upper three rows of images show leaves, and the lower three rows show root images.

**Figure 6 antioxidants-11-00917-f006:**
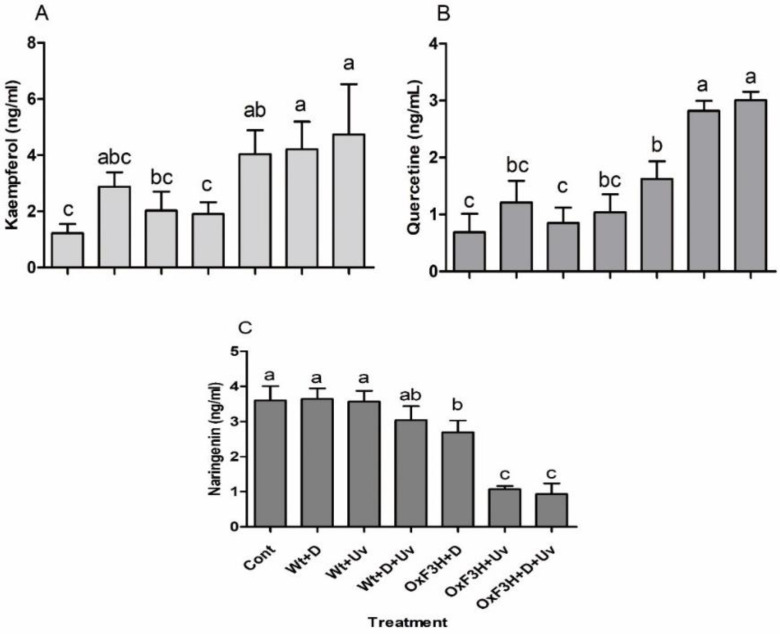
Quantification of flavonoids in wild and transgenic rice plants in response to drought and UV-B radiation stresses. (**A**) Kaempferol, (**B**) quercetin, and (**C**) naringenin accumulation level. Each data point is the mean of three replicates. Error bars represent standard errors. The bars shown with different letters are significantly different from each other as evaluated by DMRT analysis.

**Figure 7 antioxidants-11-00917-f007:**
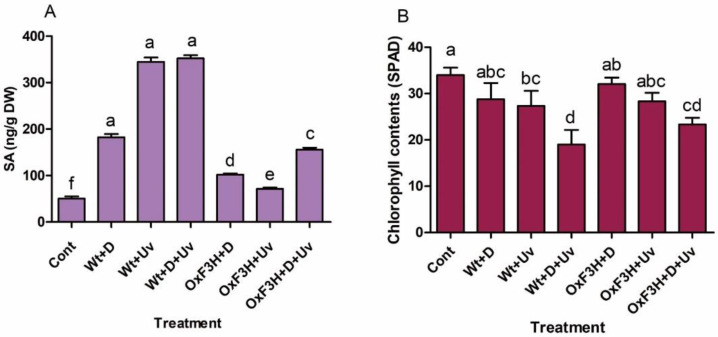
Salicylic acid and chlorophyll content are differentially regulated in wild and transgenic rice plants during drought and UV-B radiation stresses. (**A**) Salicylic acid and (**B**) chlorophyll content accumulation in wild and transgenic plants. Each data point is the mean of three replicates. Error bars represent standard errors. The bars shown with different letters are significantly different from each other as evaluated by DMRT analysis.

**Figure 8 antioxidants-11-00917-f008:**
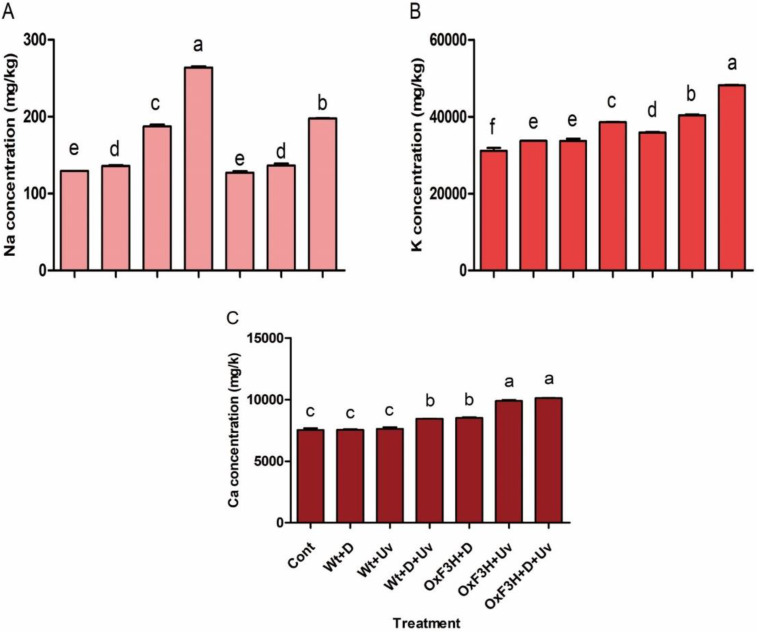
Regulation pattern of Na+, K+, and Ca+ ions in wild and transgenic rice plants under drought and UV-B radiation stress. (**A**–**C**) Accumulation of Na+, K+, and Ca+, respectively. Each data point is the mean of three replicates. Error bars represent standard errors. The bars shown with different letters are significantly different from each other as evaluated by DMRT analysis.

## Data Availability

Data is contained within the article and Appendix A.

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
