# Peer review of "Drought and UV Radiation Stress Tolerance in Rice Is Improved by Overaccumulation of Non-Enzymatic Antioxidant Flavonoids"

_antioxidants, 2022, doi:10.3390/antiox11050917_

Round 1
Reviewer 1 Report
I reviewed the manuscript "Drought and UV radiation stress tolerance in rice is improved 2by overaccumulation of non-enzymatic antioxidant flavonoids" submitted to the journal Antioxidants. Experiment seems properly conducted but the manuscript is poorly prepared and such us not worthy of publishing. Looks like someone's dissertation and it is too long without focusing on important parts. This should be significantly improved. Some of my other comments are as below:Abstract should be shortened, there is no need to mention which methods authors use, for example "Oxidative activity assays using 3,3′-diaminobenzidine (DAB) and trypan blue and glutathione pe- 26roxidase and lipid peroxidation clearly showed that... " may be shortened to " "Oxidative activity assays clearly showed that...."Similarly, the introduction is too wide, especially the part about UV irradiation. There is a lot of information about different plant species, but not about the rice which was model plants in this study. Introduction should be shortened and more focused on a manuscript topic.L 132-139. Authors did not mention which plant model they used... nor in introduction, nor in MM section. They call plants just plants. Looks like some generic part which they use for all papers. 2.6. Glutathione peroxidase activity and measurement of malondialdehyde content - both of these methods are well known and authors should explain experiments, but for sure what they presented here is too long. especially Table 1 is unnecessary.Figure 4 is too small and blurryIn the Discussion section authors repeat some facts from introduction. Here they need to be more focused on discussing their own results, not telling the story of what flavonoids are without adequate references. Again this part is unnecessary, long and hard to follow because authors repeat some things which they presented in the results section. Here authors need to compare results with other authors, especially those on the same model plants because different plants may differently react to stress.
Conclusions should be separate sections and more focused
Author Response
I reviewed the manuscript "Drought and UV radiation stress tolerance in rice is improved 2by overaccumulation of non-enzymatic antioxidant flavonoids" submitted to the journal Antioxidants. Experiment seems properly conducted but the manuscript is poorly prepared and such us not worthy of publishing. Looks like someone's dissertation and it is too long without focusing on important parts. This should be significantly improved. Some of my other comments are as below:
Reply: We are thankful for the time and effort that you dedicated to providing your valuable feedback on our manuscript. We found your comments helpful for improving our manuscript. We carefully revised our manuscript according to your valuable comments and suggestion and we hope it will now fulfill the quality of the journal and easily understandable to the readers.
Abstract should be shortened, there is no need to mention which methods authors use, for example "Oxidative activity assays using 3,3′-diaminobenzidine (DAB) and trypan blue and glutathione pe- 26roxidase and lipid peroxidation clearly showed that... " may be shortened to " "Oxidative activity assays clearly showed that....
Reply: Thank you for you valuable suggestion, we revised the abstract according to your suggestion.
"Similarly, the introduction is too wide, especially the part about UV irradiation. There is a lot of information about different plant species, but not about the rice which was model plants in this study. Introduction should be shortened and more focused on a manuscript topic.
Reply: We really appreciated your valuable comment. We intensively revised the introduction according to your suggestion. We deleted the very common informations and included others, where needed.
L 132-139. Authors did not mention which plant model they used... nor in introduction, nor in MM section. They call plants just plants. Looks like some generic part which they use for all papers.
Reply: Thank you for highlighting the important issue, we mentioned the plant name throughout the manuscript where it was needed.
2.6. Glutathione peroxidase activity and measurement of malondialdehyde content - both of these methods are well known and authors should explain experiments, but for sure what they presented here is too long. especially Table 1 is unnecessary.
Reply: Thank you for your comment, we revised the section 2.6 according to your important suggestion, we deleted the Table 1 and deleted the reference from the text.
Figure 4 is too small and blurryIn
Reply: Thank you, we revised all the figures and improve the quality.
Discussion section authors repeat some facts from introduction. Here they need to be more focused on discussing their own results, not telling the story of what flavonoids are without adequate references. Again this part is unnecessary, long and hard to follow because authors repeat some things which they presented in the results section. Here authors need to compare results with other authors, especially those on the same model plants because different plants may differently react to stress. Conclusions should be separate sections and more focused.
Reply: Thank you for your helpful and valuable comment. We intensively revised the discussion section. We limited the repletion of results and discussed our results within the support of previously published research. We hope it will be now more concise and helpful to the readers. The conclusion was also revised and summarized separately.
Reviewer 2 Report
The present work is interesting as it documents the role of specific flavonoids in rice resilience to drought and UV light. I have some suggestions for the authors to improve it before the publication:
-English version should be highly revised (e.g., line 47 "shows" should be "show")
-in introduction, the scientific reasons underlying the synthesis of ROS in presence of UV-light and drought should be described
-why did you decide to use 10% PEG 6000? The reference the authors provided here was related to peanut plants so not rice.. please, provide more details. Why the authors did not make a screening for the identification of the correct concentration of PEG to use? Similarly for the timing of exposure to the stress.
-the elution buffer in HPLC system should be indicated and also the relative flow rate.
-why RNA analysis was performed on nucleic acids extracted from plants treated for 0, 6, 12 and 24h while H2O2 was monitored after 3 days of treatment? It is not clear.. all measurements should be carried out at the same time.
-paragraph 2.6 should be strongly summarized. Too much details are reported for these two kits.
-paragraph 2.9, more details for the instrument used in this analysis should be provided.
-all figure are of low quality. Please, replace the images with others at higher definition.
-in the discussion the authors should compare the results obtained in this work with those documented in other works.. in particular, the effect of the single stressing (or both) conditions should be confronted with similar data in the same or other species.
Author Response
The present work is interesting as it documents the role of specific flavonoids in rice resilience to drought and UV light. I have some suggestions for the authors to improve it before the publication:
-English version should be highly revised (e.g., line 47 "shows" should be "show")
Reply: Thank you. We revised the manuscript by the professional native speaker.
-in introduction, the scientific reasons underlying the synthesis of ROS in presence of UV-light and drought should be described.
Reply: Thank you for your kind suggestion; we added detail information about the reason of ROS generation under drought and UV radiation.
-why did you decide to use 10% PEG 6000? The reference the authors provided here was related to peanut plants so not rice.. please, provide more details. Why the authors did not make a screening for the identification of the correct concentration of PEG to use? Similarly for the timing of exposure to the stress.
Reply: Thank you for your important comment, we used 10% PEG based on screening but unfortunately, we did not mentioned in the manuscript. Now we provided the screening result in the supplementary figure and added further informations in the text. Also included seedling stage and time of exposure.
-the elution buffer in HPLC system should be indicated and also the relative flow rate.
Reply: Thank you for your comment, we added the flow rate and eluent used.
-why RNA analysis was performed on nucleic acids extracted from plants treated for 0, 6, 12 and 24h while H2O2 was monitored after 3 days of treatment? It is not clear.. all measurements should be carried out at the same time.
Reply: You are right we quantified the transcriptional changes at 0, 6, 12 and 24 hours while oxidative stress after 3 days. As the exposure of plant to stress condition abruptly induces its transcriptional defense mechanism so we checked the gene expression right after the exposure to the stress however oxidative damage in the leaf take more time as compare to gene expression. Therefore, we visualized the leaf damage by DAB staining after three days that the changes could appear prominently and quantify the H2O2 at the same time.
-paragraph 2.6 should be strongly summarized. Too much details are reported for these two kits.
Reply: Thank you for your comment. In the original manuscript, it was short but due to the assistant editor, we included the detailed information, which she suggested through email. We revised the paragraph again according to your comment.
-paragraph 2.9, more details for the instrument used in this analysis should be provided.
Reply: Thank you for your comment, SPAD is a portable common chlorophyll measuring device.
-all figure are of low quality. Please, replace the images with others at higher definition.
Reply: Think you for highlighting the issue related to figure. We revised the figures in a good quality, we hope this time it will fulfill the journal standard.
-in the discussion the authors should compare the results obtained in this work with those documented in other works.. in particular, the effect of the single stressing (or both) conditions should be confronted with similar data in the same or other species.
Reply: Thank you for your kind suggestion; we revised the discussion section according to your suggestion and added some relative informations, we hope now it will help the reader to easily evaluate comparatively our study with the previously reported studies.
Round 2
Reviewer 1 Report
Authors revised manuscript and improved it to some point. In material and methods, when they use the name rice for the first time they should use also latin name.
Discussion is still to wide with to many not relevant results. Authors should connect the obtained results into a meaningful whole and compaire it with the other authors and works on the same lant species.
All tabalea are not visible, maybe due to the track changes option.
Author Response
Reviewer 1 (Round 2)
Authors revised manuscript and improved it to some point. In material and methods, when they use the name rice for the first time they should use also latin name.
Reply; Thank you we added latin name of the plant.
Discussion is still to wide with to many not relevant results. Authors should connect the obtained results into a meaningful whole and compaire it with the other authors and works on the same lant species.
Reply; thank you for your kind suggestion, we tried to improve the discussion of our manuscript according to your comments. We reduced the irrelevant discussion of our results, however some results are a little bit descriptive due to make it easily understandable to the readers. Regarding second part of your comment, we already compared most of our results with the published literature on the same plant but at certain point we compared with other plant species due limited published data on the same plant. The grey color shows new changes.
All tableau are not visible, maybe due to the track changes option.
Reply; Thank you, We adjust the figure size we hope now it will be visible.